# REVISITING AUXILIARY LATENT VARIABLES IN GENERATIVE MODELS

**Dieterich Lawson***
New York University
jdl404@nyu.edu

**George Tucker***, **Bo Dai**
Google Brain
{gjt, bodai}@google.com

**Rajesh Ranganath**
New York University
rajeshr@cims.nyu.edu

## ABSTRACT

Extending models with auxiliary latent variables is a well-known technique to increase model expressivity. Bachman & Precup (2015); Naesseth et al. (2018); Cremer et al. (2017); Domke & Sheldon (2018) show that Importance Weighted Autoencoders (IWAE) (Burda et al., 2015) can be viewed as extending the variational family with auxiliary latent variables. Similarly, we show that this view encompasses many of the recent developments in variational bounds (Maddison et al., 2017; Naesseth et al., 2018; Le et al., 2017; Yin & Zhou, 2018; Molchanov et al., 2018; Sobolev & Vetrov, 2018). The success of enriching the variational family with auxiliary latent variables motivates applying the same techniques to the generative model. We develop a generative model analogous to the IWAE bound and empirically show that it outperforms the recently proposed Learned Accept/Reject Sampling algorithm (Bauer & Mnih, 2018), while being substantially easier to implement. Furthermore, we show that this generative process provides new insights on *ranking* Noise Contrastive Estimation (Jozefowicz et al., 2016; Ma & Collins, 2018) and Contrastive Predictive Coding (Oord et al., 2018).

## 1 INTRODUCTION

Deep generative models with latent variables have seen a resurgence due to the influential work by Kingma & Welling (2013); Rezende et al. (2014) and their success at modeling data such as natural images (Rezende & Mohamed, 2015; Kingma et al., 2016; Chen et al., 2016; Gulrajani et al., 2016), speech and music time-series (Chung et al., 2015; Fraccaro et al., 2016; Krishnan et al., 2015), and video (Babaeizadeh et al., 2017; Ha & Schmidhuber, 2018; Denton & Fergus, 2018). The power of these models lies in the use of auxiliary latent variables to construct complex marginal distributions from tractable conditional distributions. While directly optimizing the marginal likelihood of latent variable models is intractable, we can instead maximize a variational lower bound on the likelihood such as the evidence lower bound (ELBO) (Jordan et al., 1999; Blei et al., 2017). The tightness of the bound is determined by the expressiveness of the variational family (Zellner, 1988).

Recently, there have been many advances in constructing tighter variational lower bounds for latent variable models (e.g., Burda et al. (2015); Maddison et al. (2017); Naesseth et al. (2018); Le et al. (2017); Yin & Zhou (2018); Molchanov et al. (2018); Sobolev & Vetrov (2018)). Each bound requires a separate derivation and evaluation, however, and the relationship between bounds is unclear.

We show that these bounds can be viewed as specific instances of auxiliary variable variational inference (Agakov & Barber, 2004; Ranganath et al., 2016; Maaløe et al., 2016). In particular, many partition function estimators can be justified from an auxiliary latent variable or extended state space view (e.g., Sequential Monte Carlo (Doucet et al., 2001), Hamiltonian Monte Carlo (Neal et al., 2011), Annealed Importance Sampling (Neal, 2001)). Viewed from this perspective, they can be embedded in the variational family as a choice of auxiliary latent variables. Based on the general results for auxiliary latent variables, this immediately gives rise to a variational lower bound with a characterization of the tightness of the bound. Furthermore, this view highlights the implicit (potentially suboptimal) choices made and exposes the reusable components that can be combined to form novel auxiliary latent variable schemes.

---

*Equal contributions.

The success of augmenting variational distributions with auxiliary latent variables motivates investigating a similar augmentation for generative models. When augmenting the variational distribution, the natural target distribution is the intractable posterior over the latent variables in the model. With the generative model, this introduces an extra degree of learnable flexibility (i.e., we can learn the unnormalized potential function). To illustrate this, we develop a latent variable model based on self-normalized importance sampling (Algorithm 1) which can be sampled from exactly and has a tractable lower bound on its log-likelihood. It interpolates between a tractable proposal distribution and an energy model. We show that this model is closely related to *ranking* NCE (Jozefowicz et al., 2016; Ma & Collins, 2018) and suggests a principled objective for training the noise distribution in NCE.

In summary, our contributions are:

1. We view recent tighter variational lower bounds through the lens of auxiliary variable variational inference, unifying their analysis and exposing sub-optimal design choices in algorithms such as IWAE.

2. We apply similar ideas to generative models, developing a new model based on self-normalized importance sampling which can be fit by maximizing a tractable lower bound on its log-likelihood.

3. We show that the new model generalizes ranking NCE (Ma & Collins, 2018) and provides a proof that the CPC objective (Oord et al., 2018) is a lower bound on mutual information.

4. We evaluate the proposed model and find it outperforms the recently developed approach in (Bauer & Mnih, 2018) despite being more computationally efficient and simpler to implement.

## 2 BACKGROUND

In this work, we consider learned probabilistic models of data $p(x)$. Latent variables $z$ allow us to construct complex distributions by defining the likelihood $p(x) = \int p(x|z)p(z) \, dz$ in terms of tractable components $p(z)$ and $p(x|z)$. While marginalizing $z$ is generally intractable, we can instead optimize a tractable lower bound on $\log p(x)$ using the identity

$$\log p(x) = \mathbb{E}_{q(z|x)} \left[ \log \frac{p(x, z)}{q(z|x)} \right] + D_{\mathrm{KL}} \left( q(z|x) || p(z|x) \right), \tag{1}$$

where $q(z|x)$ is a variational distribution and the positive $D_{\mathrm{KL}}$ term can be omitted to form a lower bound commonly referred to as the evidence lower bound (ELBO) (Jordan et al., 1999; Blei et al., 2017). The tightness of the bound is controlled by how accurately $q(z|x)$ models $p(z|x)$, so limited expressivity in the variational family can negatively impact the learned model.

### 2.1 AUXILIARY VARIABLE VARIATIONAL INFERENCE (AVVI)

Latent variables can also be used to define complex variational distributions $q$. As before, we define $q(z|x) = \int q(z|\lambda, x)q(\lambda|x)d\lambda$ in terms of tractable conditional distributions $q(z|\lambda, x)$ and $q(\lambda|x)$. Agakov & Barber (2004) show that

$$\mathbb{E}_{q(z|x)} \left[ \log \frac{p(x, z)}{q(z|x)} \right] = \mathbb{E}_{q(z,\lambda|x)} \left[ \log \frac{p(x, z)r(\lambda|z, x)}{q(z, \lambda|x)} \right] + \mathbb{E}_{q(z|x)} \left[ D_{\mathrm{KL}} \left( q(\lambda|z, x) || r(\lambda|z, x) \right) \right], \tag{2}$$

where $r(\lambda|z, x)$ is a variational distribution meant to model $q(\lambda|z, x)$, and the identity follows from the fact that $q(z|x) = \frac{q(z,\lambda|x)}{q(\lambda|z,x)}$. Similar to Eq. (1), Eq. (2) shows the gap introduced by using $r(\lambda|z, x)$ to deal with the intractability of $q(z|x)$. We can form a lower bound on the original ELBO and thus a lower bound on the log marginal by omitting the positive $D_{\mathrm{KL}}$ term.

## 2.2 MONTE CARLO OBJECTIVES

To tighten the variational bound without explicitly expanding the variational family, Burda et al. (2015) introduced the importance weighted autoencoder (IWAE) bound,

$$\mathbb{E}_{z_{1:K} \sim \prod_i q(z_i|x)} \left[ \log \left( \frac{1}{K} \sum_{i=1}^{K} \frac{p(x, z_i)}{q(z_i|x)} \right) \right] \leq \log p(x). \qquad (3)$$

The IWAE bound reduces to the ELBO when $K = 1$, is non-decreasing as $K$ increases, and converges to $\log p(x)$ as $K \to \infty$ under mild conditions (Burda et al., 2015). Mnih & Rezende (2016) developed Monte Carlo Objectives (MCOs), which extend this notion to any unbiased estimator $\hat{p}(x)$ of $p(x)$ by noting that

$$\mathbb{E} \left[ \log \hat{p}(x) \right] \leq \log p(x),$$

by Jensen's inequality. IWAE is the special case where the unbiased estimator is the $K$-sample importance sampling estimator. Maddison et al. (2017); Naesseth et al. (2018); Le et al. (2017) investigate MCOs in sequential models based on the unbiased estimator produced by Sequential Monte Carlo.

Many unbiased estimators can be justified as performing simple importance sampling on an extended state space (e.g., Hamiltonian Importance Sampling (Neal, 2005), Annealed Importance Sampling (Neal, 2001), and Sequential Monte Carlo (Doucet et al., 2001; Maddison et al., 2017)). In other words, we can define auxiliary variables $\lambda$ and distributions $q(\lambda|x), q(z|\lambda, x), r(\lambda|z, x)$ such that

$$\hat{p}(x) = \frac{p(x, z) r(\lambda|z, x)}{q(z, \lambda|x)},$$

with $z, \lambda \sim q(z, \lambda|x)$. It immediately follows that the estimator is unbiased and leads to a variational bound Eq. (2).

## 3 AUXILIARY LATENT VARIABLES IN VARIATIONAL FAMILIES

Viewing recent improvements in variational bounds as augmenting variational families with latent variables allows us to apply the tools of auxiliary variable variational inference to understand the tradeoffs and derivation of these algorithms. This unified view suggests novel bounds and reveals implicit design choices that may be sub-optimal.

### 3.1 IMPORTANCE WEIGHTED AUTO-ENCODERS (IWAE)

First, we explicitly work through an example with the IWAE bound. Bachman & Precup (2015) introduced the idea of viewing IWAE as auxiliary variable variational inference and Naesseth et al. (2018); Cremer et al. (2017); Domke & Sheldon (2018) formalized the notion. Consider the variational family defined by first sampling a set of $K$ candidate $z_i$s from a proposal distribution $\tilde{q}(z_i|x)$, and then sampling $z$ from the empirical distribution composed of atoms located at each $z_i$ and weighted proportionally to $p(x, z_i)/\tilde{q}(z_i|x)$. In this case, the auxiliary latent variables $\lambda$ are the locations of the proposal samples $z_{1:K}$ and the index of the selected sample, $i$.

Explicitly, let $w_i = p(x, z_i)/\tilde{q}(z_i|x)$. Then choosing the generalized densities of $q$ and $r$ as

$$q(z, z_{1:K}, i|x) = \left( \prod_{k=1}^{K} \tilde{q}(z_k|x) \right) \frac{w_i}{\sum_{k=1}^{K} w_k} \delta_{z_i}(z) \qquad (4)$$

$$r(z_{1:K}, i|z, x) = \frac{1}{K} \delta_{z_i}(z) \prod_{j \neq i} \tilde{q}(z_j|x) \qquad (5)$$

yields the IWAE bound Eq. (3) when plugged into to Eq. (2) (see Appendix A for details).

From Eq. (2), it is clear that IWAE is a lower bound on the standard ELBO for $q(z|x)$ and the gap is due to $D_{\mathrm{KL}}(q(z_{1:K}, i|z, x) || r(z_{1:K}, i|z, x))$. The choice of $r(z_{1:K}, i|z, x)$ in Eq. (5) was for

convenience and is suboptimal. The optimal choice of $r$ is

$$q(z_{1:K}, i|z, x) = q(i|z, x)q(z_{1:K}|i, z, x)$$
$$= \frac{1}{K}\delta_{z_i}(z)q(z_{-i}|i, z, x).$$

Compared to the optimal choice, Eq. (5) makes the approximation $q(z_{-i}|i, z, x) \approx \prod_{j\neq i} \tilde{q}(z_j|x)$ which ignores the influence of $z$ on $z_{-i}$ and the fact that $z_{-i}$ are not independent given $z$. A simple extension could be to learn a factored variational distribution conditional on $z$

$$r(z_{1:k}, i|z, x) = \frac{1}{K}\delta_{z_i}(z)\prod_{j\neq i} r(z_j|z, x).$$

Learning such an $r$ could improve the tightness of the bound, and we plan to explore this in future work.

## 3.2 EXTENDED STATE SPACES

More generally, many of the recent improvements in variational bounds (e.g., (Maddison et al., 2017; Naesseth et al., 2018; Le et al., 2017; Yin & Zhou, 2018; Molchanov et al., 2018; Sobolev & Vetrov, 2018)) can be viewed as importance sampling on an extended state space. By making the choice of $r$ explicit, the gap between the bound and the ELBO bound with the marginalized variational distribution is clear and this can reveal novel choices for $r$.

## 4 AUXILIARY LATENT VARIABLES IN GENERATIVE MODELS

In Section 3.1, we showed how IWAE uses self-normalized importance sampling to expand the family of $q$. Analogously, we can develop a generative model based on self-normalized importance sampling. This model draws samples from a proposal $\pi(x)$, weights them according to a potential function $U(x)$, and then draws a sample from the empirical distribution formed by the weighted samples. We define the self-normalized importance sampling (SNIS) generative process in Algorithm 1 and denote the density of the process by $p_{SNIS}(x)$. The marginal log-likelihood, $\log p_{SNIS}(x)$, can be lower bounded as

$$\log p_{SNIS}(x) \geq \mathbb{E}_{x_{2:K}} \log \left[ \frac{\pi(x)w(x)}{\frac{1}{K}\left(\sum_{j=2}^{K} w(x_j) + w(x)\right)} \right], \qquad (6)$$

for details see Appendix B. To summarize, $p_{SNIS}(x)$ can be sampled from exactly and has a tractable lower bound on its log-likelihood.

As $K \to \infty$, $p_{SNIS}(x)$ becomes proportional to $\pi(x)\exp(U(x))$. For finite $K$, $p_{SNIS}(x)$ interpolates between the tractable $\pi(x)$ and the energy model $\pi(x)\exp(U(x))$. Interestingly, $\log \pi(x)$ only shows up once in the lower bound, and simply lower-bounding it still gives a lower bound on $\log p_{SNIS}(x)$. This expands the class of allowable distributions for the proposal $\pi$ to include Variational Autoencoders (VAEs) (Kingma & Welling, 2013; Rezende et al., 2014).

To train the SNIS generative model, we can perform stochastic gradient ascent on Eq. (6) with respect to the parameters of the proposal distribution $\pi$ and the potential function $U$. When the data

---

**Algorithm 1** SNIS$(\pi, U)$ generative process

**Require:** Proposal distribution $\pi(x)$ and potential function $U(x)$.
 1: **for** $k = 1, \ldots, K$ **do**
 2:     Sample $x_k \sim \pi(x)$.
 3:     Compute $w(x_k) = \exp(U(x_k))$.
 4: **end for**
 5: Compute $\hat{Z} = \sum_{k=1}^{K} w(x_k)$
 6: Sample $i \sim \text{Categorical}(w(x_1)/\hat{Z}, \ldots, w(x_K)/\hat{Z})$.
 7: **return** $x = x_i$.

---

$x$ are continuous, reparameterization gradients can be used to estimate the gradients to the proposal distribution (Rezende et al., 2014; Kingma & Welling, 2013). When the data are discrete, score function gradient estimators such as REINFORCE (Williams, 1992) or relaxed gradient estimators such as the Gumbel-Softmax (Maddison et al., 2016; Jang et al., 2016) can be used.

Simple importance sampling scales poorly to high dimensions, so it is natural to consider augmenting the generative model with latent variables from Hamiltonian Monte Carlo or more complex samplers. We are currently exploring this.

### 4.1 CONNECTION WITH RANKING NCE AND CPC

Equation (6) is closely connected with the *ranking* NCE loss (Ma & Collins, 2018), a popular objective for training energy based models. In fact, if we consider $\pi(x)$ as our noise distribution $p_N(x)$ and set $U(x) = \tilde{U}(x) - \log p_N(x)$, then up to a constant, we recover the ranking NCE loss. The ranking NCE loss is motivated by the fact that it is a consistent objective for any $K > 1$ when the true data distribution is in our model family. As a result, it is straightforward to adapt the consistency proof from (Ma & Collins, 2018) to our setting. Furthermore, our perspective gives a coherent objective for jointly learning the noise distribution and the potential function and shows that the ranking NCE loss can be viewed as a lower bound on the log likelihood of a well-specified model regardless of whether the true data distribution is in our model family.

Moreover, this distribution provides a novel perspective on Contrastive Predictive Coding (Oord et al., 2018), a recent approach to bounding mutual information for representation learning. Starting from the well-known variational bound on mutual information due to Barber & Agakov (2003)

$$I(X, Y) = \mathbb{E}_{p(x,y)} \left[ \log \frac{p(x, y)}{p(x)p(y)} \right] \geq \mathbb{E}_{p(x,y)} \left[ \log \frac{q(x|y)}{p(x)} \right]$$

for a variational distribution $q(x|y)$, we can use the self-normalized importance sampling distribution and choose the proposal to be $p(x)$ (i.e., $p_{SNIS(p,U)}$). Applying the bound in Eq. (6), we have

$$
\begin{aligned}
I(X, Y) \quad &\geq \quad \mathbb{E}_{p(x,y)} \left[ \log \frac{p_{SNIS(p,U)}(x|y)}{p(x)} \right] \\
&\geq \quad \mathbb{E}_{p(x,y)} \mathbb{E}_{x_{2:K}} \log \left[ \frac{\exp\left(U(x, y)\right)}{\frac{1}{K} \left( \sum_j \exp\left(U(x_j, y)\right) + \exp\left(U(x, y)\right) \right)} \right].
\end{aligned}
$$

This recovers the CPC bound and proves that it is indeed a lower bound on mutual information whereas the justification in the original paper relied on approximations.

## 5 EXPERIMENTS

We evaluated generative models based on self-normalized importance sampling (SNIS) on a small, synthetic dataset as well as the MNIST dataset. To provide a competitive baseline, we use the recently developed Learned Accept/Reject Sampling (LARS) model (Bauer & Mnih, 2018). LARS trains a proposal distribution and an acceptance function (analogous to our potential function), which are used to perform rejection sampling. The output of the rejection sampling process is the generated sample. Such a process is attractive because unbiased gradients of its log likelihood can be easily computed without knowing the normalizing constant.

To ensure a sample can be generated in finite time, LARS truncates the rejection sampling after a set number of steps. Unfortunately, this change requires estimating a normalizing constant. In practice, Bauer & Mnih (2018) estimate the normalizing constant using 1024 samples during training and $10^{10}$ samples during evaluation. Even so, LARS requires additional implementation tricks (e.g., evaluating the target density, using an exponential moving average to estimate the normalizing constant) to ensure successful training, which complicate the implementation and analysis of the algorithm. On the other hand, SNIS is well-specified and has a tractable log likelihood lower bound for any $K$. As a result, no implementation tricks are necessary to train SNIS models. Moreover, SNIS weights and uses all samples instead of choosing a single sample, which we expect to be advantageous.

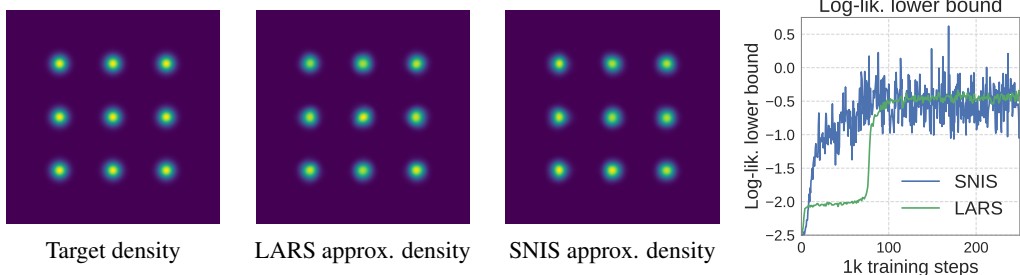

Figure 1: **Comparing the performance of LARS and SNIS on synthetic data.** Both LARS and SNIS achieve comparable data log-likelihood lower bounds, but SNIS does so much faster than LARS. The results for LARS match previously-reported results in (Bauer & Mnih, 2018). Densities plotted for LARS and SNIS are the proposal density times the exponentiated potential function evaluated at points in $[-2, 2]^2$ and approximately normalized.

| Method | Static MNIST | Dynamic MNIST |
|---:|:---:|:---:|
| VAE w/Gaussian prior | $-89.13 \pm 0.06$ | $-84.84 \pm 0.03$ |
| VAE w/SNIS prior | $\mathbf{-86.41 \pm 0.05}$ | $\mathbf{-82.67 \pm 0.02}$ |
| VAE w/LARS prior | $-86.53$ | $-83.03$ |
| SNIS w/VAE proposal | $-87.53 \pm 0.04$ | $\mathbf{-83.40 \pm 0.02}$ |
| LARS w/VAE proposal | — | $-83.63$ |

Table 1: **Comparing the performance of LARS and SNIS on MNIST.** We report 1000 sample IWAE log-likelihood lower bounds computed on the test set. SNIS numbers are the average of 5 runs, reported along with the standard deviation. LARS results are copied from Bauer & Mnih (2018).

## 5.1 SYNTHETIC DATA

As a preliminary experiment, we reproduce the synthetic data experiment from (Bauer & Mnih, 2018) which models a mixture of Gaussian densities. The target distribution is a mixture of 9 equally-weighted Gaussian densities with variance 0.01 and means $(x, y) \in \{-1, 0, 1\}^2$. Both LARS and SNIS used a fixed 2-D $\mathcal{N}(0, 1)$ proposal distribution and a learned acceptance/potential function $U(x)$ parameterized by a neural network with 2 hidden layers of size 20 and tanh activations. For both methods the number of proposal samples drawn, $K$, was set to 128. We used batch sizes of 128 and ADAM (Kingma & Ba, 2014) with a learning rate of $3 \times 10^{-4}$ to fit the models.

We plot the resulting densities and log-likelihood lower bounds in Fig. 1. As expected, SNIS quickly converges to the solution, and the potential function learns to cut out the mass between the mixture modes.

## 5.2 MNIST

Next, we evaluated SNIS on modeling the MNIST handwritten digit dataset (LeCun, 1998). MNIST digits can be either statically or dynamically binarized — for the statically binarized dataset we used the binarization from (Salakhutdinov & Murray, 2008), and for the dynamically binarized dataset we sampled images from Bernoulli distributions with probabilities equal to the continuous values of the images in the original MNIST dataset.

We tested two different model configurations: a VAE with an SNIS prior, and an SNIS model with a VAE proposal. In the first case, the SNIS prior had a Gaussian proposal distribution, and in the second case, the VAE proposal had a Gaussian prior. We chose hyperparameters to match the MNIST experiments in Bauer & Mnih (2018). Specifically, we parameterized the SNIS potential function by a neural network with two hidden layers of size 100 and tanh activations, and parameterized the VAE observation model by neural networks with two layers of 300 units and tanh activations. The latent spaces of the VAEs were 50-dimensional, and SNIS's $K$ was set to 1024. We also lin-

early annealed the weight of the KL term in the ELBO from 0 to 1 over the first $1 \times 10^5$ steps and dropped the learning rate from $3 \times 10^{-4}$ to $1 \times 10^{-4}$ on step $1 \times 10^6$. All models were trained with ADAM (Kingma & Ba, 2014).

In the SNIS model with VAE proposal, we originally used the Straight-Through Gumbel estimator (Jang et al., 2016) to estimate gradients through the discrete samples proposed by the VAE, but found that method performed worse than ignoring those gradients altogether. We suspect that this may be due to bias in the gradients. Thus, for the SNIS model with VAE proposal, we report numbers on training runs which ignore those gradients, and we plan to investigate unbiased gradient estimators in future work.

We summarize log-likelihood lower bounds on the test set in Table 1. We found that SNIS outperformed LARS even though it used only 1024 samples for training and evaluation, whereas LARS used 1024 samples during training and $10^{10}$ samples for evaluation.

## 6    DISCUSSION

In this paper, we viewed recent work on improving variational bounds through the lens of auxiliary variable variational inference. This perspective allowed us to expose suboptimal choices in existing algorithms such as IWAE, unify analysis of other methods such as ranking NCE and CPC, and derive new methods for generative modeling such as SNIS. We plan to further develop this view by embedding methods such as Hamiltonian Importance Sampling and Annealed Importance Sampling in generative models which we expect to scale better with dimension of the data space.

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

# APPENDICES

## A  IWAE BOUND AS AVVI PROOF SKETCH

We provide a sketch of a proof that the IWAE bound can be interpreted as auxiliary variable variational inference by choosing specific values for $q$ and $r$. Recall the auxiliary variable variational inference bound,

$$\mathbb{E}_{q(z|x)}\left[\log\frac{p(x,z)}{q(z|x)}\right] \geq \mathbb{E}_{q(z,\lambda|x)}\left[\log\frac{p(x,z)r(\lambda|z,x)}{q(z,\lambda|x)}\right]. \tag{7}$$

Let $q$ and $r$ be

$$q(z, z_{1:K}, i|x) = \left(\prod_{k=1}^{K}\tilde{q}(z_k|x)\right)\frac{w_i}{\sum_{k=1}^{K}w_k}\delta_{z_i}(z) \tag{8}$$

$$r(z_{1:K}, i|z, x) = \frac{1}{K}\delta_{z_i}(z)\prod_{j\neq i}\tilde{q}(z_j|x). \tag{9}$$

Then, plugging Eqs. (8) and (9) into Eq. (7) with $\lambda = (z_{1:K}, i)$ gives

$$
\begin{aligned}
\log p(x) &\geq \mathbb{E}_{q(z,\lambda|x)} \left[ \log \frac{p(x,z)r(\lambda|z,x)}{q(z,\lambda|x)} \right] \\
&= \mathbb{E}_{q(\lambda|x)} \left[ \log \frac{p(x,z_i)\frac{1}{K}\prod_{j\neq i}\tilde{q}(z_j|x)}{\frac{w_i}{\sum_j w_j}\prod_j \tilde{q}(z_j|x)} \right] \\
&= \mathbb{E}_{q(z_{1:K},i|x)} \left[ \log \frac{1}{K}\sum_j w_j \right] \\
&= \mathbb{E}_{\prod_j \tilde{q}(z_j)} \left[ \log \frac{1}{K}\sum_j w_j \right],
\end{aligned}
$$

which is the IWAE bound.

## B  DERIVATION OF LOWER BOUND ON SNIS DENSITY

We denote the density of the SNIS sampling process Algorithm 1 as $p_{SNIS}(x)$. Starting from the definition of $p_{SNIS}(x)$, we obtain Eq. (6) as

$$
\begin{aligned}
\log p_{SNIS}(x) &= \log \sum_{i=1}^{K} \int p_{SNIS}(x, x_{1:K}, i)\, dx_{1:K} \\
&= \log \sum_{i=1}^{K} \int \delta_{x_i}(x) \frac{w(x_i)}{\sum_{j=1}^{K} w(x_j)} \prod_{j=1}^{K} \pi(x_j)\, dx_{1:K} \\
&= \log \sum_{i=1}^{K} \int \frac{\pi(x)w(x)}{\sum_{j\neq i}^{K} w(x_j) + w(x)} \prod_{j\neq i} \pi(x_j)\, dx_{-i} \\
&= \log \sum_{i=1}^{K} \mathbb{E}_{x_{-i}} \left[ \frac{\pi(x)w(x)}{\sum_{j\neq i}^{K} w(x_j) + w(x)} \right] \\
&= \log \mathbb{E}_{x_{2:K}} \left[ \frac{\pi(x)w(x)}{\frac{1}{K}\left(\sum_{j=2}^{K} w(x_j) + w(x)\right)} \right] \\
&\geq \mathbb{E}_{x_{2:K}} \log \left[ \frac{\pi(x)w(x)}{\frac{1}{K}\left(\sum_{j=2}^{K} w(x_j) + w(x)\right)} \right].
\end{aligned}
$$

