# OpenReview forum: "Revisiting Auxiliary Latent Variables in Generative Models"
_ICLR.cc/2019/Workshop/DeepGenStruct — DeepGenStruct 2019_

### Official Review · AnonReviewer1 · 2019-04-16
**An interesting view of some recent work on improving variational bounds**

**Rating:** 3
**Confidence:** 2

**Review:**

This paper provides a different view of some recent work on improving variational bounds through auxiliary latent variable models. This connection gives some new insights in the existing work, e.g., IWAE, ranking NCE, and CPC. The paper also explores the possibility of using auxiliary latent variable models in the generative model.

The research in this paper is still in its early stage and it would be interesting to see how some of the unanswered questions in the current paper can be addressed.

---

### Official Review · AnonReviewer2 · 2019-04-19

**Rating:** 4
**Confidence:** 2

**Review:**

General:

This paper revisits auxiliary latent variable formulation of variational inference. Inspired by that, the authors develop a generative model based on self-normalized importance sampling (SNIS), and connect it to recent approaches such as  NCE and CPC. The view is very interesting. In experiments on MNIST, SNIS combined with VAE framework outperforms recently proposed LARS, while being faster and computationally cheaper.

Pros:

+ This paper provides a unified view of variational lower bound through auxiliary latent variables (this is not new though), relates that to the generative model side, and proposes a self-normalized importance sampling process as a generative model. This new method called SNIS can be connected with NCE and CPC. As mentioned in the paper, a unified view over different approaches might provide insights for future research

+ While only evaluated in the VAE context, the method can be potentially general and effective for other settings (as mentioned in the LARS paper).

Cons:
- I would like to see more experiments under different settings to show efficacy of the method

---

### Decision · Program_Chairs · 2019-04-22
**Acceptance Decision**

Accept